# Evoked potentials as biomarkers of hereditary spastic paraplegias: A case-control study

**Samanta Ferraresi Brighente[1], Paul Vicuña [1], Ana Luiza Rodrigues Louzada[2], Gabriela Marchisio Giordani[1], Helena Fussiger[1], Marco Antonnio Rocha dos Santos[1], Diana Maria Cubillos-Arcila[1], Pablo Brea Winckler[1,3], Jonas Alex Morales Saute [1,2,3,4] ***

**1** Graduate Program in Medicine: Medical Sciences, Universidade Federal do Rio Grande do Sul, Porto Alegre, Brazil, **2** Neurology Service, Hospital de Clínicas de Porto Alegre, Porto Alegre, Brazil, **3** Medical Genetics Service, Hospital de Clínicas de Porto Alegre (HCPA), Porto Alegre, Brazil, **4** Department of Internal Medicine, Universidade Federal do Rio Grande do Sul, Porto Alegre, Brazil

☯ These authors contributed equally to this work.
* jsaute@hcpa.edu.br

**Data Availability Statement:** All relevant data are within the paper and its Supporting Information files.

## Abstract

### Introduction

The Hereditary Spastic Paraplegias (HSP) are a group of genetic diseases that lead to slow deterioration of locomotion. Clinical scales seem to have low sensitivity in detecting disease progression, making the search for additional biomarkers a paramount task. This study aims to evaluate the role of evoked potentials (EPs) as disease biomarkers of HSPs.

### Methods

A single center cross-sectional case-control study was performed, in which 18 individuals with genetic diagnosis of HSP and 21 healthy controls were evaluated. Motor evoked potentials (MEP) obtained with transcranial magnetic stimulation and somatosensory evoked potentials (SSEP) were performed in lower (LL) and upper limbs (UL).

### Results

Central motor conduction time in lower limbs (CMCT-LL) was prolonged in HSP subjects, with marked reductions in MEP-LL amplitudes when compared to the control group (p<0.001 for both comparisons). CMCT-UL was 3.59ms (95% CI: 0.73 to 6.46; p = 0.015) prolonged and MEP-UL amplitudes were reduced (p = 0.008) in the HSP group. SSEP-LL latencies were prolonged in HSP subjects when compared to controls (p<0.001), with no statistically significant differences for upper limbs (p = 0.147). SSEP-UL and SSEP-LL latencies presented moderate to strong correlations with age at onset (Rho = 0.613, p = 0.012) and disease duration (Rho = 0.835, p<0.001), respectively. Similar results were obtained for the SPG4 subgroups of patients.

### Conclusion

Motor and somatosensory evoked potentials can adequately differentiate HSP individuals from controls. MEP were severely affected in HSP subjects and SSEP-LL latencies were

**Funding:** Fundo de Incentivo à Pesquisa e Eventos-Hospital de Clínicas de Porto Alegre (FIPE-HCPA) (Grant Number: 2019-0081). Conselho Nacional de Desenvolvimento Científico e Tecnológico (Grant Number: 303158/2020-4). The funders had no role in study design, data collection and analysis, decision to publish, or preparation of the manuscript.

**Competing interests:** The authors have declared that no competing interests exist.

prolonged, with longer latencies being related to more severe disease. Future longitudinal studies should address if SSEP is a sensitive disease progression biomarker for HSP.

## Introduction

The Hereditary Spastic Paraplegias (HSP) are a group of monogenic neurodegenerative diseases with great clinical and genetic heterogeneity, currently with 83 different *loci*[1] [1]. HSPs are rare diseases with prevalence estimations ranging from 2 to 9.1 per 100,000 individuals [2, 3]. The main features of these conditions are related to the retrograde degeneration of the longest axons of the corticospinal tract and the posterior columns [4].

HSP are classified in pure and complex forms [5]. An isolated pyramidal syndrome that predominantly affects the lower limbs, accompanied or not by neurogenic bladder and impaired vibratory sensation, characterizes "pure" HSP. Additional involvement of other systems (cognitive impairment, ataxia, parkinsonism, visual or auditory disorders, peripheral neuropathy, etc.) defines "complex" forms. Symptoms progress slowly, starting from childhood to late adulthood [4, 6], and complex forms are generally associated to a more severe disease course [2].

Although the natural history of HSPs is largely unknown, the available studies point to a very slow progression [2, 7] suggesting that clinical scales based on neurological examination might not present enough sensitivity to change for detecting disease progression, making the search for additional biomarkers a paramount task. Abnormalities in motor and somatosensory evoked potentials were previously described in HSPs, pointing neurophysiological measurements of the integrity of central motor and sensory pathways as candidate biomarkers of these diseases. However, results on motor evoked potentials (MEP) and somatosensory evoked potentials (SSEP) were heterogenous across reports, with most studies evaluating small samples or presenting poor genetic and clinical characterization [8–10].

Therefore, the aim of the present study was to characterize the role of MEP and SSEP in lower (LL) and upper limbs (UL) as biomarkers of HSPs, and to advance in the understanding of the pathophysiology of these disorders; especially concerning the involvement of central sensory pathways.

## Methods

A single center exploratory cross-sectional case-control study was performed, in which a convenience sample of 18 individuals (from 11 families) with genetic diagnosis of HSP (12 SPG4, 3 SPG5, 1 SPG7, 1 SPG11 and 1 cerebrotendinous xanthomatosis, CTX) and 21 healthy controls were evaluated. Participants were included in the study from October 2019 to February 2021. The study was approved by the Ethics in Research Committee of *Hospital de Clínicas de Porto Alegre* (GPPG-HCPA 2019–0081), Porto Alegre, Brazil. All participants were verbally informed about the conditions of the study, and signed a written consent form. In the case of children under 18 years of age, the parents signed the consent. The control group was composed of family members unrelated to the cases, such as spouses, and individuals from the community of Porto Alegre.

Eligibility for cases were previous molecular diagnosis of HSP and acceptance in participating in the study. The single subject with CTX presented a complex form of HSP, details on this case are available elsewhere [11]. Healthy subjects, unrelated, but with similar sex and ages to cases, without previous diagnosis of neurological or systemic diseases associated to motor or

sensory abnormalities were recruited as the control group. Considering the exploratory design of the study, no single primary outcome was defined and sample size estimations were not performed.

Data regarding sex, age at last examination, age at onset (first motor sign), disease duration and history of peripheral neuropathy were collected from patients and relatives or retrieved from electronic medical records. Severity of disease was evaluated with the Spastic Paraplegia Rating Scale (SPRS, range: 0–52, crescent in severity) [12]. We also analyzed motor-SPRS (mSPRS), excluding items related to pain and sphincter control (range: 0–44).

## Electrophysiological procedures

MEPs were measured to the first dorsal interosseus and *tibialis anterior* muscles after muscle activation. MEPs were obtained by single pulse transcranial magnetic stimulation with the Neuro-MS Paired Monophasic Transcranial Magnetic Stimulator (Neurosoft, Russia) device, in which an eight-shaped magnetic stimulating coil was placed over the motor cortex (total motor conduction time, TMCT) of the dominant hemisphere (C3 or C4, based on 10–20 EEG system), orientating the coil at 45% degrees from C3-C4 positions to nasion, over the seventh cervical vertebra for UL and over the fifth lumbar vertebra for LL (peripheral motor conduction time, PMCT). The pulse intensity started at the motor threshold value and increased up to about 20% of this threshold, single pulses were delivered with a frequency of 1 Hz. Ten MEPs were recorded and their amplitudes and latencies were averaged. The recording sensitivity was 100μV and 5ms per division and the filter for lower and higher frequencies was 5Hz and 10kHz, being analyzed during 100ms. Central motor conduction time (CMCT) was obtained with the direct method, by subtracting TMCT from PMCTs. When CMCT was absent, a ceiling value of 100ms was imputed. MEP amplitudes were measured from baseline to peak.

SSEPs were obtained using Neuropack M1 MEB-9200 (Nihon Kohden, Japan), in which the stimulus was generated through electrical pulses of 0.2ms applied 3 times/sec with intensities ranging from 2 to 20mV applied in medial malleolus and wrists, over the median and posterior tibial nerves respectively. On average 200 to 250 potentials were performed and superimposed to check for the reproducibility of the stimulus. Central recording electrodes were placed on the scalp over the primary sensitive area (Fz, Cz, C3, C4) with peripheral check points at the Erb point for the UL and popliteal fossa for the LL. The recording sensitivity was 2μV and 5ms per division and the filter for lower and higher frequencies was 10-2500Hz, being analyzed during 100ms. The N20 peak latencies was considered for the UL and the N50 peak latencies was considered for the LL. All neurophysiological evaluations were performed by the same evaluator (SFB), in order to reduce measurement bias. Examples of MEP and SSEP recording are presented in **S1 Fig**.

## Statistical analysis

Statistical tests were selected according to the distribution of data given by Shapiro-Wilk test and histograms. Age, age at onset, disease duration, SPRS scores and CMCT-UL presented normal distributions and were presented as means and standard deviations. The other continuous variables in the study exhibited a non-parametric distribution and were shown as median and interquartile ranges. Comparisons between cases and controls for continuous variables were performed by Mann-Whitney U-test for CMCT and for non-parametric variables and by two-tailed unpaired Student's t-test for parametric variables and by Fisher's Exact Test for categorical variables. CMCT and the SSEP latencies were considered prolonged in a given subject when the values exceeded 2 standard deviations above the mean value for the control group of

**Table 1. Main demographics characteristics of the study sample.**

| | Healthy Controls | HSP | SPG4 | p-value |
|---|---|---|---|---|
| | n = 21 | n = 18 | n = 12 | |
| Female sex | 12/21 (57.1%) | 7/18 (38%) | 5/12 (41%) | 0.341[1] |
| Age (years) | 35.2 (10.4) | 39.7 (18.7) | 38.17 (6.25) | ns[2] |
| Height (meters) | 1.66 (0.1) | 1.64 (0.17) | 1.61 (0.18) | ns[2] |
| Age at Onset (years) | - | 23.06 (15.9) | 22.83 (5,47) | - |
| Disease duration (years) | - | 16.6 (9.07) | 15.33 (2.74) | - |
| SPRS | - | 19.9 (10.65) | 17.25 (3.02) | - |
| SPRS motor | - | 16.8 (8.9) | 15.17 (2.74) | - |

Data are shown as mean and standard deviation. **HSP**: Hereditary spastic paraplegias; **ns**, not statistically significant; **SPRS**: Spastic Paraplegia Rating Scale; **mSPRS**: Motor Spastic Paraplegia Rating Scale.

[1] Fisher's Exact Test.

[2] two-tailed unpaired Student's t-test comparing HSP and controls and SPG4 subgroup and controls.

the study. Correlations were performed with Spearman correlation test for CMCT and for non-parametric variables. Statistical significance was defined as p<0.05.

## Results

The main demographic characteristics of the sample are summarized in **Table 1** and the main motor and somatosensory evoked potentials findings are described in **Table 2**. Among all the 18 individuals with HSP, only the subject with CTX had complains about decreased pain sensation in the LL. All datasets can be found in S3 Table.

### Motor evoked potentials

MEP amplitudes in UL and LL were decreased in HSP subjects when compared to healthy controls (p = 0.008, **Fig 1A**; p<0.0001, **Fig 1C**; respectively). Similar results were found in the SPG4 subgroup for amplitudes of MEP-LL (p = 0.001, **Fig 2C**), but not for amplitudes of MEP-UL, which presented a trend for reduced amplitudes when compared to healthy controls (p = 0.053, **Fig 2A**). No statistically significant correlations of MEP amplitudes for the overall

**Table 2. Main motor and somatosensory evoked potentials findings.**

| | Healthy Controls | HSP overall | p-value[1] | SPG4 | p-value[2] |
|---|---|---|---|---|---|
| | n = 21 | n = 18 | | n = 12 | |
| CMCT–UL (ms)[1] | 9.76 (0.87) | 13.36 (6.42) | 0.015 | 9.14 (3.69) | 0.81 |
| CMCT–LL (ms) | 18.11 (4.67) | 100 (63.51) | <0.001 | 69.0 (37.21) | <0.001 |
| MEP amplitude-UL (µV) | 318.4 (464.71) | 86.64 (235.03) | 0.008 | 90.54 (585.95) | 0.53 |
| MEP amplitude-LL (mV) | 100.6 (86.1) | 0 (0) | <0.001 | 0 (47.92) | <0.001 |
| SEP UL (ms) | 18.8 (2.52) | 20.17 (7.09) | 0.147 | 19.60 (6.31) | 0.59 |
| SEP LL (ms) | 45 (4.10) | 72.1 (27.2) | <0.001 | 72.1 (36.0) | <0.001 |

Data are shown as median and interquartile range except for [1]CMCT–UL. which is shown as mean and standard deviation. **CMCT**: Central Motor Conduction Time; **HSP**: Hereditary spastic paraplegias; **LL**: lower limbs; **MEP**: motor evoked potential; **ms**: milliseconds; **mV**: millivolt; **SSEP**: Somatosensory Evoked Potential; **UL**: upper limbs; **µV**: microvolt.

[1] comparisons between the overall HSP and the control group

[2] comparisons between the SPG4 subgroup and the control group

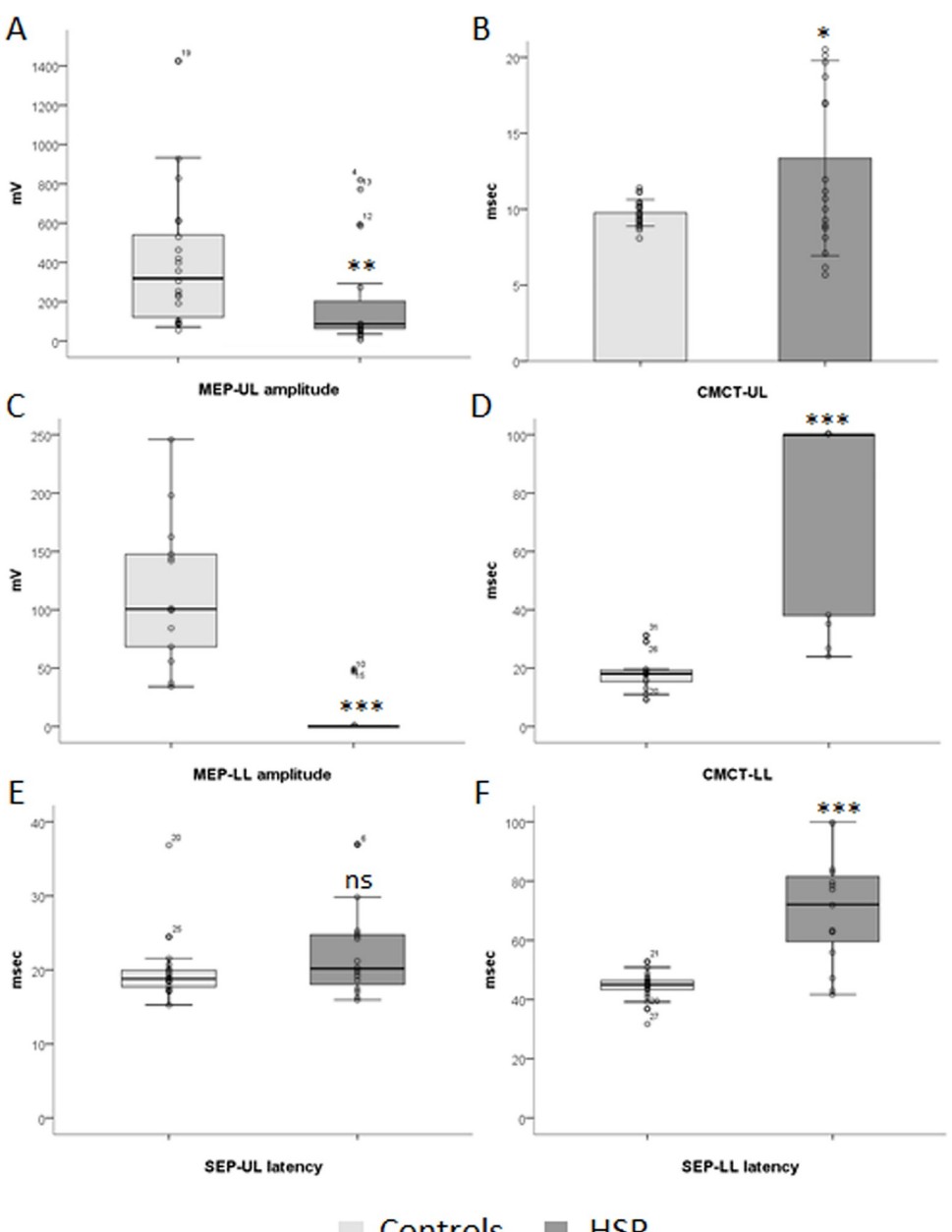

**Fig 1. Evoked potentials abnormalities in hereditary spastic paraplegias. CMCT:** Central Motor Conduction Time; **HSP:** Hereditary spastic paraplegia; **LL:** lower limbs; **MEP:** motor evoked potential; **msec**: milliseconds; **mV**: millivolt; **SSEP:** Somatosensory Evoked Potential; **UL:** upper limbs; **μV**: microvolt. *p<0.05; **p<0.01; ***p<0.001.

HSPs or SPG4 subgroup were found with disease severity variables (**S1** and **S2 Tables** respectively)

CMCT in lower limbs (LL) was strikingly different when compared to healthy controls (p<0.001, **Fig 1D**). CMCT-UL was 3.59ms (95% CI: 0.73 to 6.46; p = 0.015, **Fig 1B**) longer in HSP subjects when compared to healthy controls. Similar results were found for the SPG4 subgroup for CMCT-LL (p<0.001, **Fig 2D**), but not for CMCT-UL (p = 0.813, **Fig 2B**) when compared to healthy controls. No statistically significant correlations of CMTCs for the overall

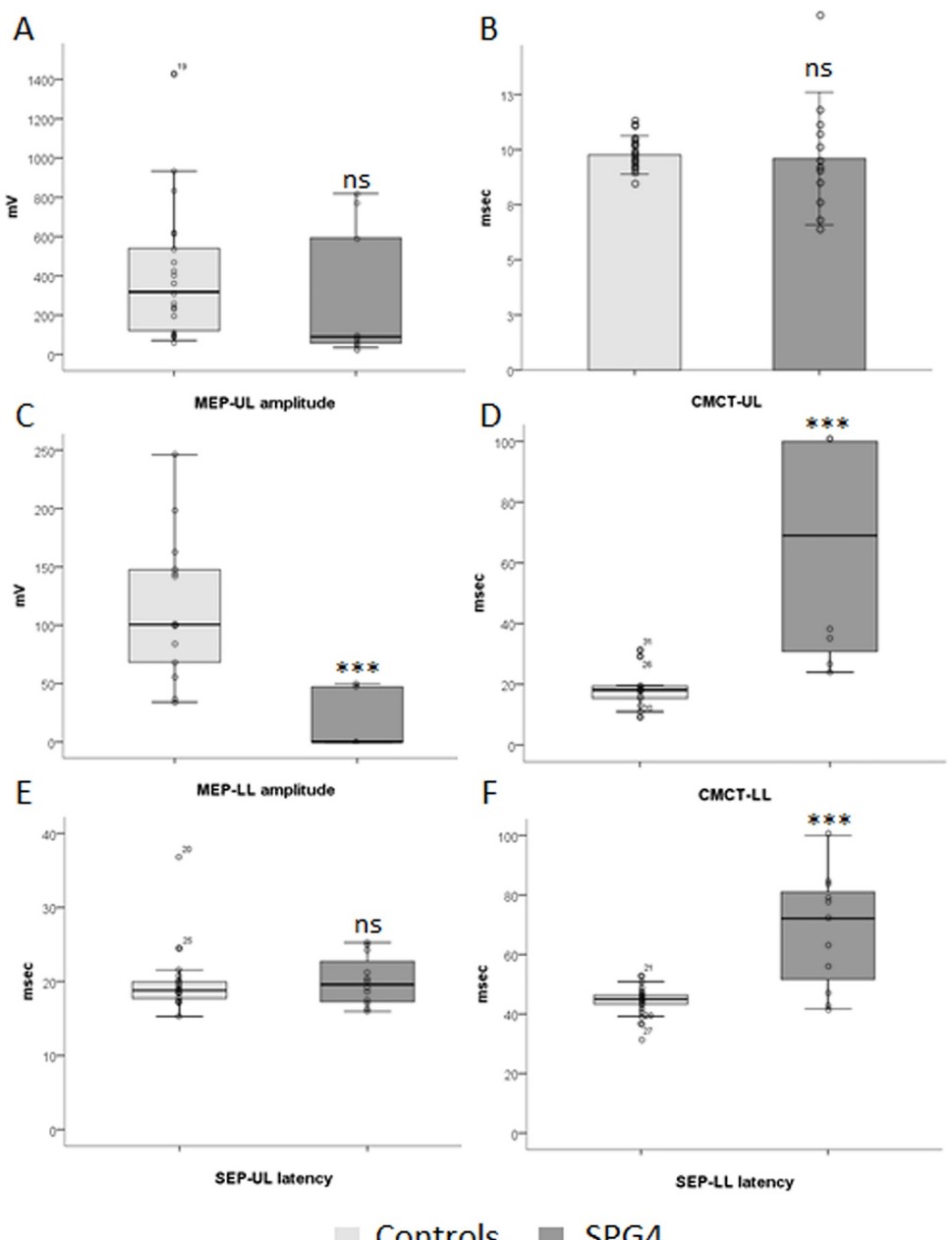

**Fig 2. Evoked potentials abnormalities in SPG4. CMCT:** Central Motor Conduction Time; **HSP:** Hereditary spastic paraplegia; **LL:** lower limbs; **MEP:** motor evoked potential; **msec:** milliseconds; **mV:** millivolt; **SSEP:** Somatosensory Evoked Potential; **UL:** upper limbs; **µV:** microvolt. ***p<0.001.

HSPs or SPG4 subgroup were found with other disease severity variables (**S1** and **S2 Tables** respectively).

## Somatosensory evoked potentials (SSEP)

Cortical SSEP latencies in lower limbs (p<0.001, **Fig 1F**), but not in upper limbs (p = 0.147, **Fig 1E**), were prolonged in HSP subjects when compared to healthy controls. SEP-LL presented a direct correlation with age (Rho = 0.628, p = 0.012) and disease duration

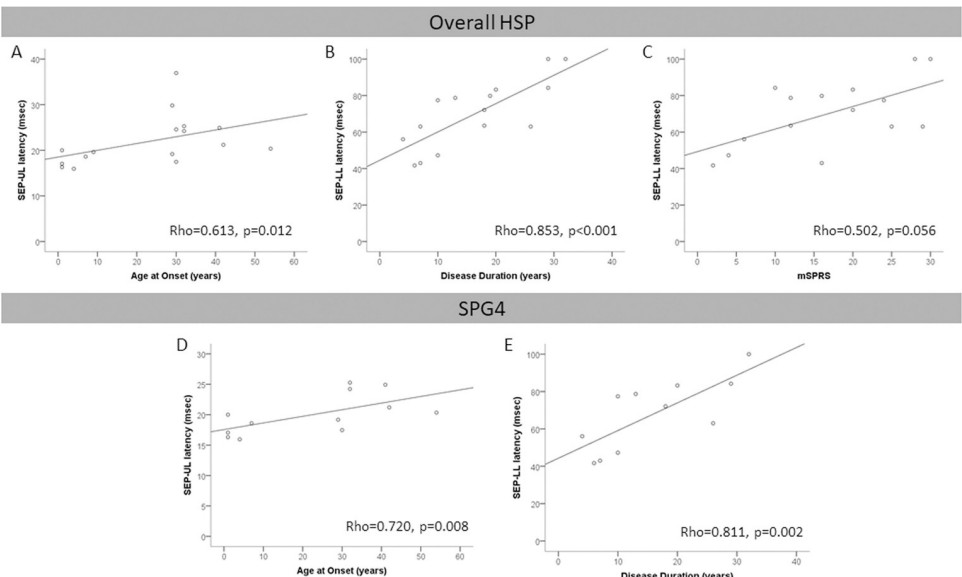

**Fig 3. Correlations of somatosensory evoked potential with disease severity variables.** LL: lower limbs; MEP: motor evoked potential; SSEP: Somatosensory Evoked Potential; SPRS: Spastic Paraplegia Rating Scale; msec: milliseconds; mSPRS: Motor Spastic Paraplegia Rating Scale; UL: upper limbs.

(Rho = 0.835, p<0.001, **Fig 3B**), and moderate, but non-significant correlations with SPRS (Rho = 0.483, p = 0.068) and motor SPRS (Rho = 0.502, p = 0.056, **Fig 3C**) in the overall HSP group. SEP-UL presented a direct correlation with age (Rho = 0.698, p = 0.003) and age at onset (Rho = 0.613, p = 0.012, **Fig 3A**) in the overall HSP group. Similar results were obtained for the SPG4 subgroups of patients, with prolonged cortical SEP latencies in lower limbs (p = 0.001, **Fig 2F**), but not in upper limbs (p = 0.593, **Fig 2E**). In the SPG4 subgroup, SEP-LL presented a direct correlation with disease duration (Rho = 0.811, p = 0.002, **Fig 3E**) and SEP-UL presented a direct correlation with age (Rho = 0.834, p = 0.001) and age at onset (Rho = 0.720, p = 0.008, **Fig 3D**). Neither SEP-UL nor SEP-LL correlate significantly with age in the control group (Rho = 0.250, p = 0.274; Rho = 0.260, p = 0.255, respectively). No statistically significant correlations of SEP latencies for the overall HSPs or SPG4 subgroup were found with other disease severity variables (**S1** and **S2** **Tables** respectively).

## Discussion

In the present study a detailed neurophysiological characterization of the integrity of central motor and sensory pathways in individuals with HSPs was performed. Our results indicated that motor and somatosensory evoked potentials can distinguish HSP subjects from healthy controls. MEPs were more severely affected in HSP subjects and SSEP-LL latencies were prolonged, with longer latencies being related to more severe disease.

Motor evoked potential presented decreased amplitudes in both the UL and the LL in the overall HSP group, but for SPG4, only MEP-LL amplitudes were reduced. Several studies have described decreased or absent MEP-LL amplitudes in HSPs [8–10, 13, 14]; however, for MEP-UL amplitudes the results from different authors were more heterogeneous, with some studies finding similar results to healthy controls [10, 13]. No significant correlations between MEP-LL and MEP-UL amplitudes with disease severity variables were found in this work, and most previous studies did not report evaluating these correlations [8–10, 13, 14].

Central motor conduction times in the lower limbs were altered in the majority HSP individuals, being absent in 9/18 (50%). In a recent systematic review of MEPs in HSPs [9], 78% of studies found abnormalities in the CMCT-LL. A study involving three centers in Germany that evaluated 128 patients with HSP, 54 of them with confirmed genetic diagnosis, reported prolongation of the CMCT-LL in 37% of the cases and absence in 36%. In the subgroup of 35 patients with SPG4, 48% presented prolonged CMCT-LL [8]. Another multicenter study carried out in Italy performed neurophysiological characterization of 49 subjects with confirmed genetic diagnosis of HSP, describing that CMCT-LL was prolonged or absent in all cases, except for one individual with SPG4 with mild phenotype. However, the correlations of CMCT findings with disease severity variables and the number of subjects who performed each of the evaluations were not clearly described for this measurement [15]. Another Italian study that evaluated 12 patients with SPG4 with an average disease duration of 20 years described prolonged CMCT-LL obtained by both the direct and indirect methods [10].

Central motor conduction times in the upper limbs were prolonged in the overall HSP group, but without differences between SPG4 and controls. Previous studies showed variable results for CMCT-UL, being abnormal in 59% of the studies reported in a recent systematic review [9]. In the multicenter studies reported above, 45% of Italian [15] and 32% (28% prolonged and 4% absent) of German patients presented prolonged CMTC-UL; however, CMCT-UL was normal in all subjects with SPG4 evaluated in the German study [8]. Eight studies reported normal CMCT-UL in HSPs, 4 of which in cohorts that included only patients with SPG4 [9, 10, 16–18].

We did not find any significant correlations between CMCTs and age at onset, disease duration and SPRS, which is similar to that found in most previous studies [9, 10, 19]. The exception was the study by Karle et al; which described a weak correlation between CMCT-LL with total SPRS and its spasticity sub-score. Another interesting data from this study was the genotype neurophysiological-phenotype correlation, in which subjects with missense variants had lower CMCT-LL latencies than subjects with truncating variants and in-frame deletions in SPAST [8]. Interestingly, such findings have been replicated in advanced neuroimaging studies in which SPG4 patients with missense variants had less severe corticospinal tract diffusivity abnormalities than patients with truncating variants in SPAST [20].

Therefore, all these data indicate important changes in CMCT in patients with HSPs, particularly when evaluated in the lower limbs. The absence of MEP in lower limbs in a significant proportion of HSP subjects indicates that this measurement has a low threshold for a ceiling effect, which may prevent the detection of relevant correlations with variables related to disease severity. In the case of SPG4, CMCT-LL results are similar to the overall HSPs, but differ for CMCT-UL, which present similar latencies to controls in most studies. Most series that evaluated the CMCT-LL, including ours, had an average disease duration close to 20 years, with moderate to severe disease according to SPRS. Due to the ceiling effect that was observed for CMCT-LL, it is unlikely that this variable will be a good biomarker of disease progression for HSP subjects with similar severity and disease durations to those reported so far; however, it will be essential that future longitudinal studies assess CMCT-LL at early disease stages, looking for an early-stage disease progression biomarker. The normal or slightly altered results of the CMCT-UL, on the other hand, suggest that future longitudinal studies should assess the progression of latencies and amplitudes of MEP in the upper limbs, seeking for an eventual role of this measure as a disease progression biomarker.

With regard to somatosensory evoked potentials, the present study found important changes only in lower limbs, in which the cortical SSEP-LL latencies were prolonged in the overall HSP and SPG4 subgroup. SSEPs are less explored in the literature, and available data are more controversial. In the German study by Karle and collaborators SSEP-UL latencies

were normal in 91% of cases, whereas SSEP-LL latencies were altered in 34% of cases, being prolonged in 27% and absent in 7% [8]. In the small Italian series that evaluated only SPG4 patients, SSEP-UL latencies were normal, whereas SSEP-LL latencies were absent in 25% of cases. However, because absent potentials data were censored, reported SSEP-LL latencies were similar to the control group [10]. In the multicenter Italian study by Martinuzzi and collaborators, cortical SSEP-LL latencies were obtained from 44 subjects with HSP, being prolonged in 30/44 (68%). SSEP-UL latencies were evaluated only in the 30 subjects with abnormal SSEP-LL, being changed in 21/30 (70%), including 9/22 (40%) of patients with SPG4 and in most patients with SPG5, SPG7 and SPG15 [15]. A congress abstract reported cortical SSEP latencies in 28 subjects with SPG4, describing changes in SSEP-UL and SEP-LL latencies in 25% and 38% of cases, respectively. The authors also reported severe temporal dispersion with decreased SSEP-LL amplitude in 61% of patients [21].

In the present study, SSEP-LL latencies strongly correlated with disease duration and moderately with age and showed trends to moderate correlations with SPRS and its motor sub score, with similar results for correlation with disease duration in the SPG4 subgroup. Despite not differing from controls, SSEP-UL latencies showed a moderate direct correlation with age at onset and with the age of subjects with HSP and with SPG4. As we did not find any correlation between age and SSEP-LL and SSEP-UL latencies in healthy controls, the correlations identified with age in HSP likely represent a confounding factor due to its high intercorrelation with age at onset and disease duration. Due to the exploratory nature of this study and its small sample size, it was not possible to perform more robust statistical analyzes to correct for this bias. The study by Karle et al. reported a correlation of SSEP-LL latencies with clinical sensory deficit [8]; however, correlations with other clinical variables were not previously described [8, 10, 15, 21].

Prolonged cortical SSEP-LL latencies in HSP are consistent with the clinical findings of vibratory sensation impairment even in pure forms of HSPs and with the results of neuroimaging studies, which indicate that the neurological changes also affect sensory pathways. Advanced neuroimaging studies using voxel-based morphometry show widespread damage to white matter in all forms of HSP and also damage to gray matter in complex forms [22]. In addition to microstructural changes in the corticospinal tract, changes in posterior brain subcortical regions were also reported in SPG4 subjects [20] as well as reduction in the cervical and dorsal spinal cord area [18, 22, 23]. Of note the reduction of spinal cord area was not accompanied by its flattening, which was interpreted as similar findings to pathological reports of SPG4 individuals, in which lateral and posterior columns involvement were the major macroscopic finding, suggesting both corticospinal tracts and posterior column involvement in this subtype [23]. *Post-mortem* neuropathological studies have also confirmed the involvement of the *gracilis* and *cuneirform* fascicles, with maximum involvement in the region of the dorsal spine [4]. Additionally, considering that dorsal root ganglion cells, which are pseudounipolar neurons, present the longest axons in humans [24] and that HSPs are one of the groups of dying-back axonopathies of long tracts, it was expected that sensory pathways abnormalities detected by SSEP would be found and would be relevant for these diseases.

Our results suggest that there are changes in SSEP-LL latencies in patients with HSP and that these latencies are correlated with longer disease durations. Future studies with larger sample sizes will be able to better detail the possible correlation with disease severity measured by SPRS. Thus, SSEP-LL latencies can be considered as disease biomarkers of HSPs and it will be essential that future longitudinal studies evaluate SSEP-LL and SSEP-UL latencies in HSPs, seeking to evaluate its role as a disease progression biomarker.

## Study limitations

The major study limitation was its small sample size and its exploratory design. Although several statistically significant differences were found for different EP measurements; we likely did not have enough power to detect smaller differences between groups and weaker to moderate correlations of MEPs and SSEPs latencies with other disease severity variables. Another study limitation is the lack of nerve conduction studies (NCS). CMCTs were obtained by the direct method, which is not influenced by motor NCS. Abnormalities in sensory NCS might have affected SSEPs; however, since only 2/18 individuals in the study presented peripheral neuropathy, both with complex HSP, and the SSEPs results of the overall HSP and the SPG4 subjects (all with pure HSP and with no evidence of peripheral neuropathy) were similar, it is unlikely that the lack of correction for sensory NCS have influenced the study results in a significant manner.

## Conclusion

Motor and somatosensory evoked potentials can distinguish HSP subjects from controls. MEP were severely affected and SSEP-LL latencies were prolonged, with longer latencies being related to more severe disease, indicating that SSEP-LL are candidate disease biomarkers for HSP. Future longitudinal studies should address if CMCT in earlier disease stages and SSEP latencies are sensitive disease progression biomarker for HSPs that could be used as surrogate outcome measures for future clinical trials.

## Supporting information

**S1 Checklist. STROBE statement—checklist of items that should be included in reports of observational studies.**
(PDF)

**S1 Fig. Examples of motor and somatosensory evoked potentials. A)** Example of a motor evoked potential (MEP); **B)** Example of a Somatosensory Evoked Potential (SSEP).
(TIF)

**S1 Table. Correlations of evoked potentials with clinical findings in the overall HSP group.** CMCT: Central Motor Conduction Time; HSP: Hereditary spastic paraplegias; LL: lower limbs; MEP: motor evoked potential; ms: milliseconds; mV: millivolt; SSEP: Somatosensory Evoked Potential; UL: upper limbs; μV: microvolt.
(DOCX)

**S2 Table. Correlations of evoked potentials with clinical findings in the SPG4 subgroup.** CMCT: Central Motor Conduction Time; HSP: Hereditary spastic paraplegias; MEP: motor evoked potential; ms: milliseconds; mV: millivolt; SSEP: Somatosensory Evoked Potential; UL: upper limbs; μV: microvolt.
(DOCX)

**S3 Table. Full dataset with clinical, genetic and neurophysiological information.** AA, Amino acid; ACMG, American College of Medical Genetics and Genomics; CMCT: Central Motor Conduction Time; LL: lower limbs; SPRS, Spastic Paraplegia Rating Scale; SSEP: Somatosensory Evoked Potential; UL: upper limbs.
(XLSX)

## Acknowledgments

We are grateful to patients for participating in this study. We also thank the Pain and Neuro-modulation Laboratory of HCPA, particularly professor Wolnei Caumo, and the Neurophysiology Unit of the Neurology Service of HCPA, particularly Ana Luiza Rodrigues Louzada, that made available the equipment for evoked potentials measurements of this study.

## Author Contributions

**Conceptualization:** Samanta Ferraresi Brighente, Pablo Brea Winckler, Jonas Alex Morales Saute.

**Data curation:** Jonas Alex Morales Saute.

**Formal analysis:** Samanta Ferraresi Brighente, Jonas Alex Morales Saute.

**Funding acquisition:** Jonas Alex Morales Saute.

**Investigation:** Samanta Ferraresi Brighente, Paul Vicuña, Gabriela Marchisio Giordani, Helena Fussiger, Marco Antonnio Rocha dos Santos, Diana Maria Cubillos-Arcila.

**Methodology:** Samanta Ferraresi Brighente, Paul Vicuña, Ana Luiza Rodrigues Louzada.

**Project administration:** Jonas Alex Morales Saute.

**Resources:** Jonas Alex Morales Saute.

**Supervision:** Jonas Alex Morales Saute.

**Validation:** Paul Vicuña, Ana Luiza Rodrigues Louzada.

**Writing – original draft:** Samanta Ferraresi Brighente, Jonas Alex Morales Saute.

**Writing – review & editing:** Paul Vicuña, Gabriela Marchisio Giordani, Helena Fussiger, Marco Antonnio Rocha dos Santos, Diana Maria Cubillos-Arcila, Pablo Brea Winckler, Jonas Alex Morales Saute.

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
