## [Decision Letter · Decision Letter 0]

10 Sep 2021

PONE-D-21-19134Evoked potentials as biomarkers of hereditary spastic paraplegiasPLOS ONE

Dear Dr. Saute,

Thank you for submitting your manuscript to PLOS ONE. After careful consideration, we feel that it has merit but does not fully meet PLOS ONE’s publication criteria as it currently stands. Therefore, we invite you to submit a revised version of the manuscript that addresses the points raised during the review process.

For acceptance, it is crucial that you adress the methodological concerns raised by reviewer 2. Especially, you should perform the additional analyses suggested by the reviewer and include a p-value correction for multiple testing in the correlation analyses. Besides, you should critically discuss the substitution of an absent TMS response with a CMCT value of 100ms.

We look forward to receiving your revised manuscript.

Kind regards,

Peter Schwenkreis

Academic Editor

PLOS ONE

“We are grateful to patients for participating in this study. The study was funded by Fundo de Incentivo à Pesquisa e Eventos-Hospital de Clínicas de Porto Alegre (FIPE-HCPA) (Grant Number: 2019-0081).

“The study was funded by Fundo de Incentivo à Pesquisa e Eventos-Hospital de Clínicas de Porto Alegre (FIPE-HCPA) (Grant Number: 2019-0081).”

Reviewers' comments:

Reviewer's Responses to Questions

**Comments to the Author**

1. Is the manuscript technically sound, and do the data support the conclusions?

Reviewer #1: Yes

Reviewer #2: Partly

2. Has the statistical analysis been performed appropriately and rigorously? 

Reviewer #1: Yes

Reviewer #2: No

3. Have the authors made all data underlying the findings in their manuscript fully available?

Reviewer #1: No

Reviewer #2: No

4. Is the manuscript presented in an intelligible fashion and written in standard English?

Reviewer #1: Yes

Reviewer #2: Yes

5. Review Comments to the Author

Reviewer #1: This is a nice study looking at evoked potentials in HSP. More data is certainly needed in this area, given this is still one of the most promising biomarkers for this condition. The paper is very well written and the methodology is complete.

However, a few comments could be made about the study which may improve it further:

- The title could perhaps better represent the contents of the study, from reading the title 'Evoked potentials as biomarkers in hereditary spastic paraplegia' it is not clear whether it is a review, systematic review or original research. The authors should consider an alternative title that is more informative such as ‘A single centre cross-sectional case-control study of evoked potentials as biomarkers of hereditary spastic paraplegia’.

- For the cerebrotendinous xanthomatosis patient to be included, the authors should clarify that this patient had a predominant HSP phenotype, since the phenotypic spectrum of this condition is variable.

- The genetic findings should be included, perhaps in a supplementary table (this would address the following point '3. Have the authors made all data underlying the findings in their manuscript fully available?').

- I thought the discussion could have been more concise and focused on the findings of the study.

- It is already known that CMCT is absent or prolonged in HSP, so the authors should make it clear what the unique contribution of there study is.

Reviewer #2: In their original article „Evoked potentials as biomarkers of hereditary spastic paraplegias”, Brighente et al. present the results of a cross sectional evoked potential study in 18 HSP patients (among these 12 SPG4) and a control cohort. As these measures are easy to perform electrophysiological biomarkers, they would be of high interest for the HSP field, and longitudinal data are still scarce. The present study, however, is limited by the low patient number, a missing disease mimic cohort (e.g., multiple sclerosis), the cross sectional design, missing p-value correction for multiple testing, and the insufficient consideration of existing studies.

Previous studies are described “poor” although some of them contained more patients with more detailed characterization than in the current work (e.g., PMID24107482 and PMID27077743). It is also claimed that “previous studies did not report evaluating these correlations [8,9,10,11,12]” although it was indeed performed in ref. [8].

Substitution of an absent TMS LL response with a value of CMCT 100ms appears arbitrary and must at least be critically discussed. With this calculation, it is not surprising that the difference for CMCT-LL is highly significant. The same is true for the SEP-LL latency.

The correlation analyses for age at onset and disease duration show significant correlations. A basic influence of age at examination itself could also explain these results. Therefore, authors should also calculate correlations to age at examination (both for the HSP and for the control cohort).

As shown in Suppl. tables 1 and 2, a high number of correlation analyses was performed and the authors highlighted the significant findings in the main text and figure 3. However, with multiple correlation analyses, p values need to be corrected for multiple testing (e.g., FDR method).

Additional issues:

Text word count on the title page is missing.

When stating “discriminatory validity” within the Abstract and in the discussion/ conclusions, authors should also perform ROC analysis and indicate AUC values. Also, the authors mention “prolonged”, but do not state reference/ cut-off values and how these were defined.

It is not reported where controls were recruited (from hospital staff, or unrelated family members, or community?).

Single data points (i.e. one point per patient/ control) should be shown in the figures.

typos: p.15 “then”, p.20 “suspicion”, p.25 “earlies”

6. PLOS authors have the option to publish the peer review history of their article (what does this mean?). If published, this will include your full peer review and any attached files.

Reviewer #1: **Yes: **Kishore Raj Kumar

Reviewer #2: No

---

## [Author Response · Author response to Decision Letter 0]

12 Oct 2021

RESPONSE TO REVIEWERS

Response: The text was checked according to the template suggested by the journal.

Response: Additional information about the consent form and control group was attached in the body of the article.

“All participants were verbally informed about the conditions of the study, and signed a written consent form. In the case of children under 18 years of age, the parents signed the document. The control group was composed of family members unrelated to the cases, such as spouses, and individuals from the community of Porto Alegre.”

3.Thank you for stating the following in the Acknowledgments Section of your manuscript:

“We are grateful to patients for participating in this study. The study was funded by Fundo de Incentivo à Pesquisa e Eventos-Hospital de Clínicas de Porto Alegre (FIPE-HCPA) (Grant Number: 2019-0081).

Response: The funding information was withdrawal from this section and provided only in the online submission form.

4 – In your Data Availability statement, you have not specified where the minimal data set underlying the results described in your manuscript can be found. PLOS defines a study's minimal data set as the underlying data used to reach the conclusions drawn in the manuscript and any additional data required to replicate the reported study findings in their entirety. All PLOS journals require that the minimal data set be made fully available. For more information about our data policy, please see http://journals.plos.org/plosone/s/data-availability.

Response: In order to attend PLOS Data policy we have now added Supplemental Table 3, an Excel file in which anonymized raw data for clinical, genetic and neurophysiological variants is provided. 

Point 5

COMENTS 

Reviewer 1

Reviewer #1: 

This is a nice study looking at evoked potentials in HSP. More data is certainly needed in this area, given this is still one of the most promising biomarkers for this condition. The paper is very well written and the methodology is complete.

However, a few comments could be made about the study which may improve it further

Response: We thank the reviewer comment

- The title could perhaps better represent the contents of the study, from reading the title 'Evoked potentials as biomarkers in hereditary spastic paraplegia' it is not clear whether it is a review, systematic review or original research. The authors should consider an alternative title that is more informative such as ‘A single centre cross-sectional case-control study of evoked potentials as biomarkers of hereditary spastic paraplegia’.

Response: We agree with the reviewer point. The title was changed to: 

“Evoked potentials as biomarkers of hereditary spastic paraplegias: a case-control study”

- For the cerebrotendinous xanthomatosis patient to be included, the authors should clarify that this patient had a predominant HSP phenotype, since the phenotypic spectrum of this condition is variable.

Response: We have added in the eligibility criteria section that the single subject with CTX presented a complex form of HSP and that details on this case are available elsewhere (Burguez et al, 2017).

- The genetic findings should be included, perhaps in a supplementary table (this would address the following point '3. Have the authors made all data underlying the findings in their manuscript fully available?').

Response: We have added Supplemental Table 3, an Excel file in which anonymized raw data for clinical, genetic and neurophysiological variants is provided. 

- I thought the discussion could have been more concise and focused on the findings of the study.

Response: We have reviewed the discussion section and performed cuts in the text in order to reduce its length. 

- It is already known that CMCT is absent or prolonged in HSP, so the authors should make it clear what the unique contribution of there study is.

Response: We agree that it is already known that CMCT is absent or very 

prolonged in HSP; however, some previous studies did not have a complete genetic characterization and there is a lack of data regarding the correlation of CMCT with other clinical severity outcome. So, we consider that the confirmation of CMCT prolongation in HSP in our population was an interesting finding. Also, the lack of correlation of CMCT with other disease severity variables may indicate that this biomarker in patients with mean disease duration of around 15 years is too much compromised, probably presenting a ceiling effect that hamper significant correlations with other disease severity biomarkers like clinical scales and disease duration. It will be interesting to evaluate CMCT in HSP patients in the first years of clinical presentation, indicating that future studies with CMCT in HSP are needed.

Nevertheless, the most relevant finding in this sample was the data found on sensory latencies. The sensory system is generally not considered a core system involved in the pathophysiology of HSP and we have found differences from controls and significant correlations with disease duration and age at onset, and a trend for a moderate direct correlation with SPRS severity. Perhaps the fact that the sensory system remains more preserved than the motor makes it a better disease biomarker during intermediate or late stages of the disease. 

We have added in the conclusion that SSEP-LL are candidate disease biomarkers for HSP, trying to emphasize this major result. 

 

Reviewer 2

In their original article “Evoked potentials as biomarkers of hereditary spastic paraplegias”, Brighente et al. present the results of a cross sectional evoked potential study in 18 HSP patients (among these 12 SPG4) and a control cohort. As these measures are easy to perform electrophysiological biomarkers, they would be of high interest for the HSP field, and longitudinal data are still scarce. 

Response: We thank the reviewer comment and we agree his/her points. 

The present study, however, is limited by the low patient number, a missing disease mimic cohort (e.g., multiple sclerosis), the cross sectional design, missing p-value correction for multiple testing, and the insufficient consideration of existing studies.

Response: We thank the reviewer comment. We will address the comments point by point bellow

Low patient number: We agree with this point, which was stated in the study limitations sections on discussion; however, considering the disease rarity and the number of subjects in previous studies with HSP and neurophysiological characterization we consider that our study provides significant addition to scientific literature. We have recruited all subjects from a reference center in a state of 10 million inhabitants in Southern Brazil with genetic confirmation of HSP diagnosis. Some patients didn´t agree to participate due to the pandemic of COVID-19. For sure, future multicenter studies will be needed to achieve larger samples sizes and robust conclusions; however, these future studies should focus on the clues given by studies like ours, as studying CMCT in early stages of HSP and to assess longitudinally SSEP to see if it is also a disease progression biomarker. 

A missing disease mimic cohort (e.g., multiple sclerosis): Considering that we were not searching for a diagnostic test, as HSP are genetic conditions and the final genetic diagnosis is achieved by the gold standard (genetic testing) we did not consider that a disease mimic group would be necessary for the study aims. We changed the term discriminatory validity to discriminate case and healthy controls to avoid any misunderstanding regarding this point. Importantly, a given disease biomarker does not need to be specific of HSP. The main intention here was to assess if the biomarker was different from healthy controls, which meant that it is associated with the disease state and to assess if patients with greater severities indicated by different variables (eg: disease duration, SPRS, age at onset) presented worse results than patients with milder disease, trying to search for disease severity biomarkers. 

Cross sectional design: Indeed, this design does not allow us to asses the properties of MEP and SSEP as disease progression biomarkers. Only longitudinal studies will be able to do so. We are following these subjects and we intend to report the longitudinal findings after 24 months in a future publication. 

Missing p-value correction for multiple testing: As disclosed by the reviewer the study has a small sample and it is also exploratory, so we didn´t perform correction for multiple testing. We will discuss this issue in more details bellow. 

insufficient consideration of existing studies: we have performed a systematized review of the literature for this project before it began and we have cited the main studies that were found. Of note, the systematic review by Siow et al summarized many of the studies regarding MEP and we have cited this comprehensive review. 

Previous studies are described “poor” although some of them contained more patients with more detailed characterization than in the current work (e.g., PMID24107482 and PMID27077743). It is also claimed that “previous studies did not report evaluating these correlations [8,9,10,11,12]” although it was indeed performed in ref. [8].

Response: Some of the previous works have larger number of patients, like the ones cited by the reviewer; however, even with a larger sample, the study PMID27077743 had broader objectives and the neurophysiological characterization lacks a lot of details. On the other hand, the study by Karle et al is very well described, being a multicenter German study with a larger number of subjects, 128. As we have stated in the discussion section, the study by Karle et al, along with the others cited papers did not presented data on the amplitudes of MEP, indeed they have focused on CMCT findings. In the discussion section, when we are describing CMCT results we have detailed the findings of the study by Karle et al and we highlight such parts bellow:

“A study involving three centers in Germany that evaluated 128 patients with HSP-suspicion, 54 of them with confirmed genetic diagnosis, reported prolongation of the CMCT-LL in 37% of the cases and absence in 36% . In the subgroup of 35 patients with SPG4, 48% presented prolonged CMCT-LL [8].

…The exception was the study by Karle et al., which described a weak correlation between CMCT-LL with total SPRS and its spasticity sub-score. Another interesting data from this study was the genotype neurophysiological-phenotype correlation, in which subjects with missense variants had lower CMCT-LL latencies than subjects with truncating variants and in-frame deletions in SPAST [8].”

Substitution of an absent TMS LL response with a value of CMCT 100ms appears arbitrary and must at least be critically discussed. With this calculation, it is not surprising that the difference for CMCT-LL is highly significant. The same is true for the SEP-LL latency.

Response: The use of the ceiling value of 100ms for CMCT in absent TMS LL response was indeed arbitrary and intended to avoid the exclusion of cases with the greatest severity on this biomarker. However, the chosen value had no impact on the obtained p-values, since for both between groups comparisons and for correlations involving CMCT and SSEP we have used ranked non-parametric tests (Mann-Whitney U-test and Spearman) in which the rank, but the not the raw value is used to calculate p-values and Rho. We have added the statistical method for evaluating CMCT more clearly in the statistical analysis, to avoid misinterpretations. We have provided data always as median and interquartile ranges, except for CMCT-UL which did not have a ceiling effect and in which the data was normally distributed. 

The correlation analyses for age at onset and disease duration show significant correlations. A basic influence of age at examination itself could also explain these results. Therefore, authors should also calculate correlations to age at examination (both for the HSP and for the control cohort).

Response: We consider that the raised issues were already addressed during the paper. In the results section we have informed that “SEP-LL presented a direct correlation with age (Rho=0.628, p=0.012)… in the overall HSP group” …” In the SPG4 subgroup…SEP-UL presented a direct correlation with age (Rho=0.834, p=0.001). We have also detailed that “Neither SEP-UL nor SEP-LL correlate significantly with age in the control group (Rho=0.250, p=0.274; Rho=0.260, p=0.255, respectively).”. 

In the discussion section we have described that: “Despite not differing from controls, SSEP-UL latencies showed a moderate direct correlation with age at onset and with the age of subjects with HSP and with SPG4. As we did not find any correlation between age and SSEP-LL and SSEP-UL latencies in healthy controls, the correlations identified with age in HSP likely represent a confounding factor due to its high intercorrelation with age at onset and disease duration. Due to the exploratory nature of this study and its small sample size, it was not possible to perform more robust statistical analyzes to correct for this bias.”

As shown in Suppl. tables 1 and 2, a high number of correlation analyses was performed and the authors highlighted the significant findings in the main text and figure 3. However, with multiple correlation analyses, p values need to be corrected for multiple testing (e.g., FDR method).

Response: The study sample size is small and it has an exploratory nature, therefore we did not perform corrections for multiple comparisons. The exploratory design is depicted in the methods section and also in the limitations section. 

After the reviewer comment, we have performed the FDR method and only the SSEP-LL correlation with disease duration remained significant at a Benjamini-Hochberg Adjusted P value of 0.024. Considering that the results related to SPRS and age at onset (moderate correlations) should be confirmed in future multicenter studies, with greater study power, we decided to keep our original statistical design and we have reported the results without corrections. We reinforce the exploratory nature of the results to support our decision. Being too restrictive in small sample size studies will likely increase Type II error, which is not the objective of an exploratory study like ours. 

Additional issues:

Text word count on the title page is missing.

Response: We added word for title and abstract. Word count for the manuscript is not required according to the journal policies. We would add it anyway; however, because Tables should appear inside the text of the main file, as well as legends, we did not add this information. 

When stating “discriminatory validity” within the Abstract and in the discussion/ conclusions, authors should also perform ROC analysis and indicate AUC values. 

Response: We changed the term “discriminatory validity” to “the given variable discriminate case and healthy controls” to avoid any misunderstanding regarding this point. Considering that we didn´t aim to evaluate the diagnostic properties of MEP e SSPE, ROC curves were not calculated. 

Also, the authors mention “prolonged”, but do not state reference/ cut-off values and how these were defined.

Response: We considered that normality values for neurophysiological variables taken from books or guidelines would be less reliable than a comparison with the local population, so we decided to evaluate a healthy control group. In the present version we have avoided the term prolonged, but we also have described that: “CMCT and the SSEP latencies were considered prolonged in a given subject when the values exceeded 2 standard deviations above the mean value for the control group of the study” in the methods section. 

It is not reported where controls were recruited (from hospital staff, or unrelated family members, or community?).

Response: The control group was composed of family members unrelated to the cases, such as spouses, and individuals from the community of Porto Alegre. We have added this in the methods section. 

Single data points (i.e. one point per patient/ control) should be shown in the figures.

Response: Done. Figures were updated accordingly. 

typos: p.15 “then”, p.20 “suspicion”, p.25 “earlies”

Response: Corrected. Thanks.

---

## [Editor Report · Decision Letter 1]

19 Oct 2021

Evoked potentials as biomarkers of hereditary spastic paraplegias: a case-control study

PONE-D-21-19134R1

Dear Dr. Saute,

We’re pleased to inform you that your manuscript has been judged scientifically suitable for publication and will be formally accepted for publication once it meets all outstanding technical requirements.

Kind regards,

Peter Schwenkreis

Academic Editor

PLOS ONE
---

## [Editor Report · Acceptance letter]

25 Oct 2021

PONE-D-21-19134R1 

Evoked potentials as biomarkers of hereditary spastic paraplegias: a case-control study 

Dear Dr. Saute:

I'm pleased to inform you that your manuscript has been deemed suitable for publication in PLOS ONE. Congratulations! Your manuscript is now with our production department. 

Kind regards, 

on behalf of

Dr. Peter Schwenkreis 

Academic Editor

PLOS ONE